# Shortcutting Pre-trained Flow Matching Diffusion Models is Almost Free Lunch

Xu Cai[*†]    Yang Wu[‡]    Qianli Chen[*]    Haoran Wu[*]    Lichuan Xiang[§]    Hongkai Wen[§]

## Abstract

We present an ultra-efficient post-training method for shortcutting large-scale pre-trained flow matching diffusion models into efficient few-step samplers, enabled by novel *velocity field self-distillation*. While shortcutting in flow matching, originally introduced by shortcut models, offers flexible trajectory-skipping capabilities, it requires a specialized step-size embedding incompatible with existing models unless retraining from scratch—a process nearly as costly as pretraining itself.

Our key contribution is thus imparting a more aggressive shortcut mechanism to standard flow matching models (e.g., Flux), leveraging a unique distillation principle that obviates the need for step-size embedding. Working on the velocity field rather than sample space and learning rapidly from self-guided distillation in an online manner, our approach trains efficiently, e.g., producing a 3-step Flux <1 A100 day. Beyond distillation, our method can be incorporated into the pretraining stage itself, yielding models that inherently learn efficient, few-step flows without compromising quality. This capability also enables, to our knowledge, the first *few-shot* distillation method (e.g., 10 text-image pairs) for dozen-billion-parameter diffusion models, delivering state-of-the-art performance at almost free cost.

## 1 Introduction

Recent advancements in accelerating diffusion models Song et al. [2021b], Ho et al. [2020] have significantly reduced the number of denoising steps required for sample generation without compromising quality. For instance, while DDPM Ho et al. [2020] typically requires 1000 sampling steps, DDIM Song et al. [2021a] achieves a substantial speed-up with as few as 50 steps. Nevertheless, as pre-trained diffusion models Podell et al. [2024], Esser et al. [2024], BlackForestLabs [2024] continue to scale to (tens of) billions of parameters, the computational cost remains a critical bottleneck for real-world applications. Most recent diffusion models favor Flow Matching (FM) Lipman et al. [2023], also known as rectified flow Liu et al. [2023], due to its simplified Ordinary Differential Equation (ODE) solver and exact likelihood evaluation. FM achieves this by learning a velocity field that deterministically maps Gaussian noise to clean data along a (nearly-)straight flow trajectory. However, as detailed in Section 2, only a handful of existing works explicitly optimize the distillation objective specifically for flow matching models.

Among the related works, shortcut models Frans et al. [2025] stand out for their unified approach to train both multi-step and one-step diffusion with a step-size embedding, allowing the model to inherently support both modes once properly trained, as further detailed in Section 3.2. The core idea of shortcut models is to create direct shortcuts along potentially curved trajectories, by always

---

[*]The authors conducted this work as independent researchers, pursued as a hobby and without institutional affiliation.

[‡]AI Research Center, iHuman Inc..

[§]Department of Computer Science, University of Warwick.

[†]Corresponding author: `caitreex@gmail.com`.

 Project page: `shortcutfm.github.io`.

predicting the *mean* direction between two equally spaced timesteps. The challenge, however, is that the aforementioned property is missing in all existing pre-trained FM models. As a result, naively applying shortcut methods to standard flow matching would require architectural modifications or complete retraining from scratch to integrate step-size conditioning.

Motivated by this, and inspired by related works on velocity matching (refer to Section 2 for more details), we introduce a novel framework for *ShortCutting Flow Matching* (SCFM) through a highly efficient distillation paradigm. In contrast to majority existing distillation approaches, SCFM inherits the *end-to-end* training manner from shortcut models, while differing substantially in *distillation principle*, how step-size information is incorporated, and training algorithm design.

To elaborate on the high-level idea behind this distillation principle, our key insight is that an obvious obstacle to few-step sampling in flow matching models lies in the inconsistency between their theoretical formulation and empirical behavior. We ask: what if the entire non-linear velocity field between noise and data, spanning a long time horizon, could be uniformly forced to a nearly straight trajectory (e.g., via distillation)? In such cases, the explicit step-size parameter may become unnecessary, as the model would now align with the theoretical principles of rectified flow, which naturally supports efficient arbitrary-step sampling. The detailed intuition can be found in Remark 1. With this in mind, we summarize our key contributions as follows:

- We propose SCFM that operates in velocity space, enforcing linear consistency across all timesteps. This consistency is jointly derived from both the teacher and an online-inherited student through a novel dual-target distillation objective. Notably, our approach eliminates the need for explicit progressive distillation; instead, an *end-to-end* training scheme automatically straightens curved velocity trajectories. In contrast to shortcut models, our method does not rely on an explicit step-size parameter to regulate varying velocities.
- SCFM benefits significantly from self-distillation, enabling highly efficient training and entirely removing the need for massive dataset to mimic teacher models—a requirement common to most prior approaches. To demonstrate the novelty and effectiveness of *few-shot* SCFM, we validate using as few as 10 training samples, yet still achieved competitive performance. To our knowledge, this represents the first successful demonstration of few-shot distillation for large-scale diffusion models.
- Our method is designed to generalize to any pre-trained flow matching model. To validate this, we successfully distilled a 12B-parameter Flux model BlackForestLabs [2024] into a 3-step sampler within a single day on an A100 GPU, achieving SOTA performance in both quantitative scores and visual quality, without the aid of adversarial distillation Sauer et al. [2024b,a], a standard component in all baselines.

## 2   Related Work

We follow the categorization in a recent survey Fan et al. [2025], existing pre-trained diffusion distillation methods can be broadly classified based on their training objectives: output/distribution-based or trajectory-based. Output-based distillation adopts a straightforward approach: minimizing the discrepancy between the outputs of the teacher and the student. These methods typically employ sophisticated loss functions. For example, Progressive Distillation (PD) Salimans and Ho [2022] uses an $\ell_2$ loss, while other methods leverage information-theoretic losses such as KL divergence Yin et al. [2024] or Fisher divergence Zhou et al. [2024], just to name a few.

Trajectory-based distillation, on the other hand, focuses on the transformation path from noisy inputs to clean images. The pioneering work on Consistency Models (CMs) Song et al. [2023] introduced the idea of mapping any noised data point along the trajectory directly to its clean counterpart, enabling one-step diffusion. More recent approaches extend this idea by targeting near-future noisy states in the trajectory, rather than the final clean image, as the distillation target. This adjustment results in more practical and efficient training strategies for popular pre-trained models such as SDXL Podell et al. [2024]. Notable examples include Luo et al. [2023], Wang et al. [2024], Kim et al. [2024], Ren et al. [2024], Wang et al. [2025], and many more.

More recent and advanced pre-trained diffusion models, such as SD3 Esser et al. [2024] and Flux BlackForestLabs [2024], adopt flow matching (a.k.a., rectified flow) Liu et al. [2023] as their underlying trajectory, in contrast to previous approaches like EDMs Karras et al. [2022]. The notable advantage of rectified flow is its natural compatibility with few-step diffusion once properly

trained. As hinted in Section 1, in practice, however, class-conditional models often deviate from this idealized linear trajectory, especially in high-noise regions where the mapping often exhibit curvature. Consequently, most of the above mentioned distillation algorithms, which were originally developed for conventional trajectories, are not yet well suited to flow matching models.

Despite this limitation, InstaFlow Liu et al. [2024b], a successor to rectified flow Liu et al. [2023], introduces a two-stage training strategy: pre-training with the rectified flow objective followed by velocity-based distillation. What sets InstaFlow apart is its distillation is carried out in the velocity field, which contrasts sharply with strategies that rely on denoised data targets. Perhaps the most relevant work to ours is the shortcut model Frans et al. [2025], which unifies one-step and multi-step diffusion pre-training by learning in the velocity field and conditioning on step-size throughout training. Their shortcut principle motivated our approach, which leverages velocity jumping through a novel distillation targets tailored to pre-trained flow matching dynamics.

Other orthogonal approaches for achieving effective flow matching distillation include LADD Sauer et al. [2024a], a predecessor to the well-known Adversarial Diffusion Distillation (ADD) Sauer et al. [2024b], which introduces a GAN-like objective Goodfellow et al. [2014] based on deceiving a discriminator. Notably, ADD-style methods are orthogonal to most diffusion distillation techniques, as the additional adversarial loss is intended to enhance the alignment between the teacher and student. SANA-Sprint Chen et al. [2025] takes a different route by substituting the original flow matching trajectory. Instead, it adopts TrigFlow Lu and Song [2025], a generalized trajectory framework that unifies flow matching and EDM under continuous-time consistency. Their approach is then combined with LADD to further improve generation quality.

## 3 Preliminaries

### 3.1 Flow Matching Models

Given a data sample $\mathbf{x}_0 \sim \mathcal{D}$ drawn from data distribution $\mathcal{D}$, flow matching (a.k.a, rectified flow) diffusion models learn to reverse the noising process through the following continuous-time trajectory:

$$\mathbf{x}_t = (1 - t)\mathbf{x}_0 + t\mathbf{z}, \tag{1}$$

where $t \in [0, 1]$ and $\mathbf{z} \sim \mathcal{N}(0, \mathbf{I})$. The forward noising process evolves from $t = 0$ to $t = 1$, gradually perturbing $\mathbf{x}_0$ into $\mathbf{z}$. For notational convenience, we let $\mathbf{x}_1$ represent the fully noised version of $\mathbf{x}_0$. The corresponding ODE with respect to $t$ takes a simple linear form:

$$\mathbf{v}_t = \frac{\partial \mathbf{x}_t}{\partial t} = \mathbf{x}_1 - \mathbf{x}_0, \tag{2}$$

where $\mathbf{v}_t$ is referred to as the velocity. Given training pairs $\{\mathbf{x}_0, \mathbf{x}_1\}$, our objective is to predict $\hat{\mathbf{v}}_t = \mathcal{V}_{\boldsymbol{\theta}}(\mathbf{x}_t, t)$ using a neural network $\mathcal{V}_{\boldsymbol{\theta}}$ with parameters $\boldsymbol{\theta}$. We note that $\mathcal{V}_{\boldsymbol{\theta}}(\mathbf{x}_t, t)$ denotes the minimal form of the velocity predictor, while a fully parameterized version may be written as $\mathcal{V}_{\boldsymbol{\theta}}(\mathbf{x}_t, t, c, w)$, incorporating a condition $c$ (e.g., a prompt) and a Classifier-Free Guidance (CFG) scale $w$ Ho and Salimans [2021].

Since $\mathbf{x}_1$ follows from the normal distribution, the estimator $\hat{\mathbf{v}}_t = \mathbb{E}[\mathbf{v}_t \mid \mathbf{x}_t, t]$ is unbiased. Consequently, the Mean Squared Error (MSE) of the estimator $\hat{\mathbf{v}}_t$ is known to minimize the variance, meaning that the estimator with the *least variance* is considered the best in terms of MSE. Finally, the neural network $\mathcal{V}_{\boldsymbol{\theta}}$ can be optimized by minimizing the following loss function:

$$\mathcal{L}(\boldsymbol{\theta}) = \mathbb{E}_{\mathbf{x}_1 \sim \mathcal{D}, \mathbf{x}_0 \sim \mathcal{N}(0, \mathbf{I})} \left[ \left\| \mathcal{V}_{\boldsymbol{\theta}}(\mathbf{x}_t, t) - \mathbf{v}_t \right\|^2 \right] \tag{3}$$

$$= \mathbb{E}_{\mathbf{x}_1 \sim \mathcal{D}, \mathbf{x}_0 \sim \mathcal{N}(0, \mathbf{I})} \left[ \left\| \mathcal{V}_{\boldsymbol{\theta}}\big((1 - t)\mathbf{x}_0 + t\mathbf{x}_1, t\big) - (\mathbf{x}_1 - \mathbf{x}_0) \right\|^2 \right], \tag{4}$$

where (4) follows from substituting the trajectory (1) and velocity (2) definition into (3).

The inference procedure solves the ODE backward from $t = 1$ to $t = 0$. An $n$-step sampler employs $n + 1$ predefined timesteps $\{t_i\}_{i=0}^n$, where $1 = t_0 > t_1 > \cdots > t_n = 0$. Samples are then generated through $n$ iterative updates:

$$\mathbf{x}_{t_{i+1}} = \mathbf{x}_{t_i} - (t_i - t_{i+1})\mathcal{V}_{\boldsymbol{\theta}}(\mathbf{x}_{t_i}, t_i), \quad \mathbf{x}_{t_0} \sim \mathcal{N}(0, \mathbf{I}), \tag{5}$$

with final output $\mathbf{x}_{t_n}$ as the generated sample.

## 3.2 Shortcut Models

As discussed in Section 2, shortcut models $\mathcal{V}_{\boldsymbol{\theta}}(\mathbf{x}_t, t, d)$ Frans et al. [2025] generalize standard flow matching by incorporating a variable step size $d$ (e.g., $1/n$). The key property is self-consistency across different step sizes:

$$\mathbf{x}_{t+2d} = \mathbf{x}_t - 2d\mathcal{V}_{\boldsymbol{\theta}}(\mathbf{x}_t, t, 2d) \tag{6}$$

$$\approx \mathbf{x}_t - d\mathcal{V}_{\boldsymbol{\theta}}(\mathbf{x}_t, t, d) - d\mathcal{V}_{\boldsymbol{\theta}}(\mathbf{x}_{t+d}, t+d, d), \quad 0 < d \le \frac{1}{2}, \tag{7}$$

where (6) and (7) reflect the compositional property that one $2d$-step prediction equals two consecutive $d$-steps. Equating (6) and (7) and scale both sides with $\frac{1}{2d}$ leads to the self-consistency loss:

$$\mathcal{L}_{\text{sc}}(\boldsymbol{\theta}) = \left( \mathcal{V}_{\boldsymbol{\theta}}(\mathbf{x}_t, t, 2d) - \mathcal{V}_{\boldsymbol{\theta}^-}(\mathbf{x}_t, t, 2d) \right)^2, \tag{8}$$

where $\boldsymbol{\theta}^-$ denotes the *stopgrad* version of $\boldsymbol{\theta}$ (typically maintained via exponential moving average (EMA)), and the target $\mathcal{V}_{\boldsymbol{\theta}^-}(\mathbf{x}_t, t, 2d)$ is computed as:

$$\mathcal{V}_{\boldsymbol{\theta}^-}(\mathbf{x}_t, t, 2d) = \frac{1}{2}\mathcal{V}_{\boldsymbol{\theta}^-}(\mathbf{x}_t, t, d) + \frac{1}{2}\mathcal{V}_{\boldsymbol{\theta}^-}(\mathbf{x}_{t+d}, t+d, d). \tag{9}$$

Shortcut models are thus optimized through jointly minimization of:

- The flow matching loss (4) (with $d = 0$), ensuring basic generation capability.
- The self-consistency loss (8), facilitating shortcut abilities via varied step sizes.

During training, step sizes $d$ in (8) are uniformly sampled from $\{\frac{1}{n}, \frac{2}{n}, \frac{4}{n}, \dots, \frac{1}{2}\}$ to promote multi-scale consistency. To effectively encode $d$, rotary positional embedding Su et al. [2024] is employed to preserve the relative scaling relationships between different step sizes.

# 4 The Shortcut Distillation Method

Inspired by (6), (7) and (9), we propose to implicitly train the awareness of $d$ in $\mathcal{V}_{\boldsymbol{\theta}}(\mathbf{x}_t, t)$, rather than including it as an explicit variable as in Section 3.2. As hinted in Section 1, this naturally encourages the trajectory to become nearly straight, rather than merely relying on shortcuts along a curved path.

A natural adaptation would extend the framework of PD Salimans and Ho [2022] to operate on velocity fields instead of sample fields, giving rise to an iterative teacher-student distillation process. The key challenge, then, lies in preserving compatibility with standard diffusion architectures while accommodating variable step sizes. To this end, consider a sequence of $n$ time intervals $\{d_i\}_{i=1}^n$ (which may be non-uniform) satisfying

$$\sum_{i=1}^n d_i = 1, \quad d_i = t_{i-1} - t_i. \tag{10}$$

Building on the self-consistency principles from (6) and (7), we derive the velocity space consistency:

$$(d_i + d_{i+1})\mathcal{V}_{\boldsymbol{\theta}}(\mathbf{x}_{t_i}, t_i) \approx d_i\mathcal{V}_{\boldsymbol{\theta}}(\mathbf{x}_{t_i}, t_i) + d_{i+1}\mathcal{V}_{\boldsymbol{\theta}}(\mathbf{x}_{t_{i+1}}, t_{i+1}). \tag{11}$$

Rearranging terms yields the following form of our distillation target:

$$\mathcal{V}_{\boldsymbol{\theta}}(\mathbf{x}_{t_i}, t_i) = \frac{d_i}{(d_i + d_{i+1})}\mathcal{V}_{\boldsymbol{\theta}}(\mathbf{x}_{t_i}, t_i) + \frac{d_{i+1}}{(d_i + d_{i+1})}\mathcal{V}_{\boldsymbol{\theta}}(\mathbf{x}_{t_{i+1}}, t_{i+1}), \tag{12}$$

where we slightly abuse notation by denoting the left-hand side as the training target, even though it shares the same symbolic form as the intermediate outputs on the right-hand side. Similar to (9), for the target computed through the teacher and the EMA model, we replace all instances of $\boldsymbol{\theta}$ in (12) with $\boldsymbol{\theta}^*$ and $\boldsymbol{\theta}^-$ to differentiate the sources.

Equation (12) captures the weighted interpolation of velocity fields across time intervals, forming the foundation for our shortcut distillation process. Next, if one adapts the progressive distillation pipeline, the first stage would begin with fine-grained step sizes $d_i$, where $\mathcal{V}_{\boldsymbol{\theta}^*}(\mathbf{x}_{t_i}, t_i)$ represents the velocity prediction directly from the teacher model. The distillation then proceeds through stages

$\mathcal{V}_{\boldsymbol{\theta}^*} \to \mathcal{V}_{\boldsymbol{\theta}^-} \to \cdots$ with increasingly coarser-grained $d_i$. However, this naive approach may face challenges in determining optimal transition points between stages, considering the propagation and amplification of approximation errors.

Recall from Section 3.2 that shortcut models ground their learning at small step sizes ($d \to 0$) via (4) while self-bootstrapping at larger steps ($d \in \{\frac{2^i}{n}\}_{i=0}^{\lfloor \log_2 n \rfloor - 1}$) via (8). Such joint optimization is theocratically equal to progressive optimization: first training the model at a small step size, and then learning from the previous model at increasing step sizes. Motivated by these insights, we propose the *ShortCut distillation for Flow Matching* (SCFM) objective:

$$\mathcal{L}_{\mathrm{scfm}}(\boldsymbol{\theta}) = \frac{1}{N}\bigg( \sum_{i=1}^{k} \Big(\mathcal{V}_{\boldsymbol{\theta}}\big(\mathbf{x}_t, t\big) - \mathcal{V}_{\boldsymbol{\theta}^*}\big(\mathbf{x}_t, t\big)\Big)^2 + \sum_{i=k+1}^{N} \Big(\mathcal{V}_{\boldsymbol{\theta}}\big(\mathbf{x}_t, t\big) - \mathcal{V}_{\boldsymbol{\theta}^-}\big(\mathbf{x}_t, t\big)\Big)^2 \bigg), \quad (13)$$

where $\mathcal{V}_{\boldsymbol{\theta}^*}\big(\mathbf{x}_t, t\big)$ and $\mathcal{V}_{\boldsymbol{\theta}^-}\big(\mathbf{x}_t, t\big)$ are calculated through (12), $N$ is the total batch size and $0 < k < N$ controls the teacher/self-teaching mixing ratio.

**Remark 1.** The the SCFM loss $\mathcal{L}_{\mathrm{scfm}}$ can be seen as a generalization of several existing distillation methods. When $k = N$, it encompasses a broad class of existing trajectory-based distillation approaches (see Appendix B for details); setting $k = 0$ corresponds to a progressive-style distillation scheme. Analogous to shortcut learning, the first term in (13) transfers knowledge from the teacher model to a coarser student, while the second term self-distills this knowledge into finer-scale students. This enables implicit progressive self-distillation as suggested in Section 1, leading to asymptotic convergence toward a one- or few-step sampler within a single training phase.

To build intuition, consider a teacher model with $n$ diffusion steps. The first term trains a student at $\sim n/2$ steps to coarsely rectify the velocity direction. The second term, at the same time, enforces consistency between this $n/2$-step student and finer-scale versions (e.g., $n/4$ steps). By randomly sampling target steps from $\{n/4, n/8, \ldots, 1\}$ in the second term, the loss ensures self-consistency across all these finer scales. Interestingly, even if all finer-scale directions are initially wrong, they can still be rectified through the first term via teacher guidance. Consequently, the second term rapidly straightens the overall trajectory, while the first term corrects the direction w.r.t. the teacher.

### 4.1 Training Details

We update the EMA of the stop-gradient parameters $\boldsymbol{\theta}^-$ using the conventional update rule:

$$\boldsymbol{\theta}^- = \mu\boldsymbol{\theta}^- + (1 - \mu)\boldsymbol{\theta}, \quad \text{with } \mu = 0.999 \text{ by default.} \quad (14)$$

To enable efficient post-training, we adopt low-rank adaptation (LoRA) Hu et al. [2022], where the model parameters are expressed as $\boldsymbol{\theta} = \boldsymbol{\theta}_0 + \Delta\boldsymbol{\theta}$, with $\boldsymbol{\theta}_0$ denoting the frozen pre-trained weights and $\Delta\boldsymbol{\theta}$ the trainable LoRA. Let $\Delta\boldsymbol{\theta}^-$ denote the LoRA parameter corresponding to the stopgrad model. Since $\boldsymbol{\theta}^- = \boldsymbol{\theta}_0 + \Delta\boldsymbol{\theta}^-$, we can derive the EMA update rule for the LoRA parameters as:

$$\begin{aligned} \Delta\boldsymbol{\theta}^- &= \boldsymbol{\theta}^- - \boldsymbol{\theta}_0 \\ &= \mu\boldsymbol{\theta}^- + (1 - \mu)\boldsymbol{\theta} - \boldsymbol{\theta}_0 \\ &= \mu(\boldsymbol{\theta}_0 + \Delta\boldsymbol{\theta}^-) + (1 - \mu)(\boldsymbol{\theta}_0 + \Delta\boldsymbol{\theta}) - \boldsymbol{\theta}_0 \\ &= \mu\Delta\boldsymbol{\theta}^- + (1 - \mu)\Delta\boldsymbol{\theta}, \end{aligned} \quad (15)$$

where we have used both the update for $\boldsymbol{\theta}^-$ in (14) and the LoRA decomposition of $\boldsymbol{\theta}$. To further accelerate training, we may apply a cyclic restarting strategy to (15), where $\Delta\boldsymbol{\theta}^-$ is reinitialized with the current $\Delta\boldsymbol{\theta}$ every fixed number of iterations (e.g., every 1000 steps). For a more in-depth understanding, we refer readers to the essential Section 5.4, which offers key insights into the final SCFM training strategy that further improves convergence speed. The vanilla training algorithm is provided in Algorithm 1, located in Appendix A.

## 5 Experiments

In this section, we present empirical studies to evaluate SCFM. We begin by showcasing numerical and visual comparisons, followed by comprehensive ablations in Section 5.4 (including few-shot learning, detailed in Appendix F) that explores how to achieve the most efficient and performant SCFM configuration.

Table 1: Comprehensive comparison with SOTA approaches. Latency is measured with resolution $1024 \times 1024$ in BF16 precision on A100, averaged over 10 runs. The timing specifically reflects the duration spent within the transformer block loop, excluding pre/post-processing overhead. We highlight the **best** and second-best results across all metrics.

| | Methods | Steps | Latency (s) | ($\Delta$) FID ↓ | FID ↓ (w.r.t. base) | CLIP ↑ |
|---|---|---|---|---|---|---|
| **Base** | SD3.5-Large | 32 | 15.14 | 18.62 | N/A | 34.97 |
| **Distillation-based** | SD3.5L-Turbo | 8 | 1.95 | +7.03 (25.65) | 8.18 | 33.81 |
| | **SD3.5L-SCFM** | 8 | 3.71 | **+0.32** (18.94) | **2.65** | **33.91** |
| | SD3.5L-Turbo | 4 | 0.94 | +6.36 (24.98) | 6.98 | 33.03 |
| | **SD3.5L-SCFM** | 4 | 1.81 | **+4.45** (23.07) | **6.89** | **33.40** |
| | SD3.5L-Turbo | 3 | 0.71 | +6.85 (25.47) | 7.76 | 32.25 |
| | **SD3.5L-SCFM** | 3 | 1.31 | **+5.35** (23.98) | **7.41** | **32.46** |
| **Base** | Flux.1-Dev | 32 | 15.62 | 27.43 | N/A | 33.60 |
| | Flux-TeaCache-0.6 | 32 | 5.36 | -2.25 (25.18) | 2.18 | 33.24 |
| | Flux-TeaCache-0.8 | 32 | 4.33 | -3.15 (24.28) | 3.56 | 32.77 |
| **Distillation-based** | Flux-Hyper-SD | 8 | 3.71 | +1.37 (28.80) | 3.20 | 33.46 |
| | Flux-TDD | 8 | 3.71 | -0.37 (27.06) | 4.02 | 33.17 |
| | **Flux-SCFM** | 8 | 3.71 | **+0.16** (27.59) | **2.58** | **33.76** |
| | Flux-Hyper-SD | 4 | 1.80 | -0.64 (26.79) | 5.45 | 32.94 |
| | Flux-TDD | 4 | 1.80 | -2.62 (24.81) | 5.50 | 32.57 |
| | Flux-Schnell | 4 | 1.80 | -6.41 (21.02) | 6.76 | 33.17 |
| | **Flux-SCFM** | 4 | 1.80 | **-0.45** (26.98) | **4.50** | **33.20** |
| | Flux-Hyper-SD | 3 | 1.33 | -1.52 (25.91) | 9.65 | 31.95 |
| | Flux-TDD | 3 | 1.33 | -4.46 (22.97) | 8.26 | 31.38 |
| | Flux-Schnell | 3 | 1.33 | -6.58 (20.85) | 7.06 | 33.06 |
| | **Flux-SCFM** | 3 | 1.33 | **-1.01** (26.42) | **6.34** | **33.10** |

## 5.1 Training settings

We conduct our experiments on a filtered subset of the LAION dataset Schuhmann et al. [2022], specifically the LAION-POP dataset LaionPop [2024], which contains 600k* samples in total with aesthetic scores >0.5. For evaluation, we adopt the widely used COCO-30k validation set Lin et al. [2014]. All training and evaluation are performed on a single NVIDIA A100 80GB GPU.

We use two large-scale, high-capacity pre-trained flow-matching models as teachers: Flux.1 Dev (12B) BlackForestLabs [2024] and SD3.5 Large (8B) Esser et al. [2024]. Our method successfully distills a 32-step Flux teacher into a 3-step student in under 24 A100 GPU hours, demonstrating remarkable training efficiency. For comparison, progressive-style distillation methods typically require thousands of GPU hours Salimans and Ho [2022], highlighting the practical efficiency of our approach. All experiments are conducted on images resized to approximately $512 \times 512$ in area. Since our method operates in the velocity space, we observed no performance degradation when evaluating at $1024 \times 1024$ resolution. We use the AdamW optimizer Loshchilov and Hutter [2019] with a learning rate of $2e^{-5}$, a batchsize of $N=16$, and set $\frac{k}{N}=0.4$ in (13), chosen bootstrappingly.

Since Flux.1 Dev, denoted as $\mathcal{V}(\mathbf{x}, t, c, w)$, is already CFG-distilled using an embedded guidance representation, we simply randomly sample the CFG scale from the range $w \in [0, 8]$, ensuring both the student and teacher share the same $w$ for consistency. In contrast, SD3.5 does not incorporate CFG embedding, instead, we sample CFG values from the range $[3.5, 5]$ (as we aim to retain flexibility at inference time by supporting adjustable CFG scales), rather than distilling to a fixed value. As a result, our method requires a CFG-conditioned formulation for all velocity predictors $\mathcal{V}(\mathbf{x}, t, c)$ appearing in (12). As in Ho and Salimans [2021], this can be expressed as:

$$\mathcal{V}(\mathbf{x}, t, c) = \mathcal{V}(\mathbf{x}, t, c) + w\big(\mathcal{V}(\mathbf{x}, t, c) - \mathcal{V}(\mathbf{x}, t, \emptyset)\big), \tag{16}$$

where $\emptyset$ denotes the unconditional (null) prompt. During inference, we find that setting the CFG scale within the range $[4.5, 6]$ yields stable and reliable results for both models. Alternative methods

---

*This dataset is publicly available; as specified in Section 5.4, since our method converges rapidly, we used only a random subset (<50% of full dataset).

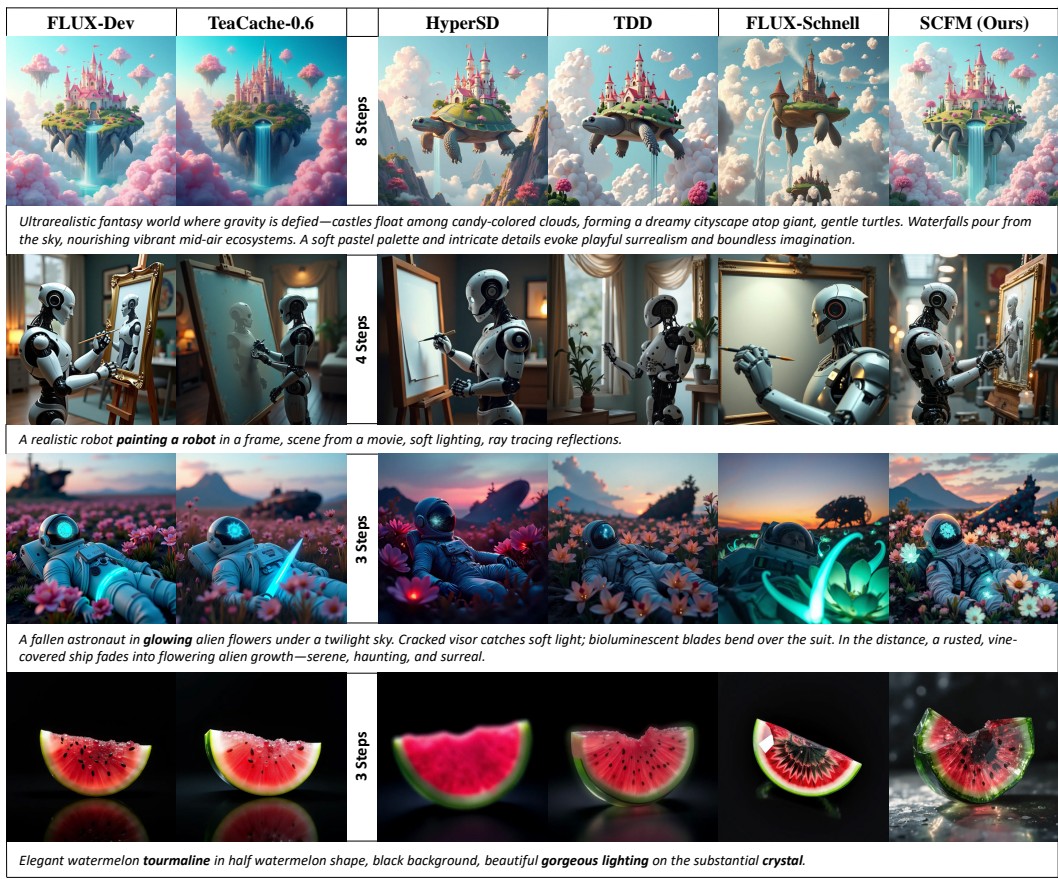

| FLUX-Dev | TeaCache-0.6 | | HyperSD | TDD | FLUX-Schnell | SCFM (Ours) |
|---|---|---|---|---|---|---|

*Ultrarealistic fantasy world where gravity is defied—castles float among candy-colored clouds, forming a dreamy cityscape atop giant, gentle turtles. Waterfalls pour from the sky, nourishing vibrant mid-air ecosystems. A soft pastel palette and intricate details evoke playful surrealism and boundless imagination.*

*A realistic robot **painting a robot** in a frame, scene from a movie, soft lighting, ray tracing reflections.*

*A fallen astronaut in **glowing** alien flowers under a twilight sky. Cracked visor catches soft light; bioluminescent blades bend over the suit. In the distance, a rusted, vine-covered ship fades into flowering alien growth—serene, haunting, and surreal.*

*Elegant watermelon **tourmaline** in half watermelon shape, black background, beautiful **gorgeous lighting** on the substantial **crystal**.*

Figure 1: Visual comparisons on Flux: samples from the original teacher and the TeaCache-accelerated variant uses 32 sampling steps.

that rely on a CFG embedding can reduce inference cost by avoiding a separate unconditional pass (thus halving runtime), but typically require architectural changes and substantially more training from near-scratch. While both approaches are expected to yield similar generation quality, we opt for the simpler, backward-compatible strategy.

## 5.2 Evaluation Metrics and Baselines

Our evaluation primarily focuses on the fidelity between the student and teacher models—specifically, how well the student shortcuts (i.e., preserves) the behavior of the teacher. Additionally, we assess the standalone quality of the distillation outputs to ensure that acceleration does not come at the cost of generating visually compelling and semantically adherence results.

To this end, we deviate from conventional FID Heusel et al. [2017] evaluation against an open reference dataset (as commonly done in prior works), instead, we propose to compute the FID between teacher and student outputs under fixed random seeds. This allows us to directly quantify how much fidelity is lost when drastically reducing the number of inference steps. This teacher-student FID serves as a better proxy for evaluating distillation fidelity, especially when the teacher itself is already a strong pre-trained model. As a complement to the similarity measurement between the student and teacher models, we evaluate the visual quality and semantic alignment of the accelerated/distilled model using CLIP Radford et al. [2021], equipped with the checkpoints from Koukounas et al. [2024] over the official OpenAI versions (both base and large), due to their improved image-text alignment (see Appendix G for detailed comparisons).

Although diffusion distillation is widely studied, to the best of our knowledge, no prior work has conducted distillation experiments on recent large-scale diffusion models like Flux or SD3.5—apart from the official releases of Flux-Schnell BlackForestLabs [2024] and SD3.5-Turbo Esser et al.

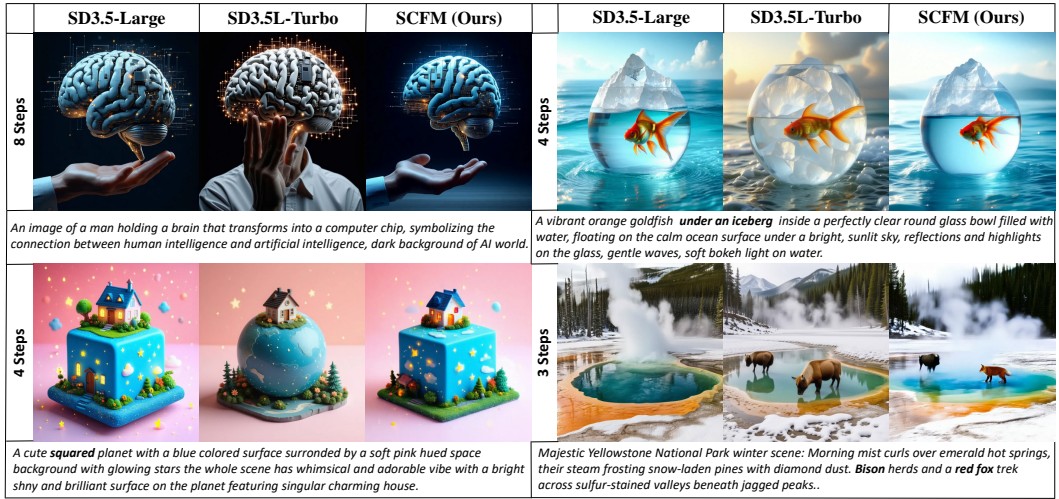

|  | SD3.5-Large | SD3.5L-Turbo | SCFM (Ours) |  | SD3.5-Large | SD3.5L-Turbo | SCFM (Ours) |

Figure 2: Visual comparisons on SD3.5-Large: samples from the original teacher uses 32 sampling steps.

[2024]. We further note that recent trajectory-based distillation methods such as HyperSD Ren et al. [2024] and TDD Wang et al. [2025] were subsequently applied to Flux (and thus were not included in the official benchmarks reported in their papers); for fair comparison, we use their publicly released checkpoint weights to establish baselines. Notably, all these baselines have utilized ADD/LADD for improved performance, whereas ours dose not. On the other hand, we also include a TeaCache Liu et al. [2024a] baseline for Flux using their publicly released code[†]. TeaCache is a training-free approach that accelerates inference by selectively skipping redundant timesteps based on output fluctuation, rather than directly reducing the number of steps. As TeaCache introduces a quality-latency trade-off via a threshold parameter, we evaluate their fastest (0.8) and second-fastest (0.6) configurations as provided in their official Flux implementation.

## 5.3 Analysis

We present our main evaluation results in Table 1, focusing on the key metrics discussed above. In particular, we adopt $\Delta$FID to quantify the distributional shift between the student and teacher models with respect to a common fixed validation dataset. To further strengthen this comparison, we also construct a *teacher-generated reference dataset* using fixed CFG values and random seeds. This enables a direct, controlled measurement of student-teacher fidelity, isolating the impact of distillation from other sources of variance. Across all three evaluation metrics—$\Delta$FID, FID, and CLIP—our proposed method consistently achieves the best performance.

As noted in Section 5.1, we did not perform CFG embedding distillation for SD3.5; as a result, our method requires twice the number of function evaluations during inference compared to CFG-distilled variants like SD3.5-Turbo. Nonetheless, our framework is fully compatible with CFG embedding distillation, which we leave for possible future exploration. While TeaCache-0.6 achieves a competitive FID score, it lags behind our method in inference speed and CLIP score. Moreover, our slowest distilled 8-step student outperforms the fastest TeaCache-0.8 in both speed and generation quality—despite requiring only a few hours of training—offering a fair and highly efficient alternative to this training-free approach.

Visual comparisons for Flux are depicted in Figure 1. Our method, SCFM, maintains the highest fidelity to the 32-step teacher, particularly in the 8-step student, while simultaneously improving prompt adherence—evident in the 3- and 4-step students. For instance, SCFM better captures details like "robot painting" and "glowing flowers". Similar trends are seen in the SD3.5 results in Figure 2, where the 3- and 4-step students accurately capture prompt-specific concepts like "squared", "bison" and "fox", sometimes better than the teacher. On the other hand, a closer examination of our few-step samples reveals noticeably improved clarity, detail, sharpness, and composition compared to other

---

[†]We omit the SD3.5 version, as TeaCache has not been extended to support this model at the time of submission, see https://github.com/ali-vilab/TeaCache.

methods. While the image quality is on par with Flux-Schnell, our approach better *preserves the characteristics* of the original Flux-Dev model, as evidenced by the lowest fidelity score and superior visual consistency with the teacher. Additional visual results are provided in Appendices H and I.

### 5.4 Further Ablations and Discussion

Given the simplicity of the vanilla SCFM introduced in Section 4, several promising avenues for further exploration naturally emerge. The following ablation studies trace the empirical motivations behind the final version of our SCFM training algorithm:

1. As an early reference to Appendix B, where we compare our method with other distillation approaches, we consider an ablation in what if we mix the teacher and stopgrad student solvers in the first term of (13). Does this impact overall performance or training speed? As shown in Appendix C, we observe slight improvement in quantitative results. This indicates that the training scheme is robust and amenable to further refinements.

2. As hinted at the end of Section 4.1, employing cyclic restarting for the stopgrad model can significantly accelerate convergence, confirmed by CLIP score progression curves in Appendix D. However, the strategy may become detrimental during very late-stage training, nevertheless, we find that an 8-step student safely converges by approximately the 2000th iteration—equivalent to around 10 A100 GPU hours when distilling Flux in our setup.

3. The above findings suggest that a rapidly accumulated stop-gradient model facilitates faster convergence. Building on this and motivated by the first point, we propose mixing teacher and stopgrad models with different EMA decay rates in (13): a "fast" model with a lower decay parameter ($\mu = 0.99$) and a "slow" model with a higher decay parameter ($\mu = 0.999$). This dual-EMA strategy eliminates the need for manual cyclic restart, instead enabling a fully automatic mechanism that leads to even faster convergence. With this setup, the 8-step student safely converges by around the 1000th iteration ($\sim$5 A100 GPU hours), and the overall model reaches 3-step performance in under 24 GPU hours. Further details are provided in Appendix E.

4. Inspired by the observations above, we note that SCFM is already highly efficient to train: by the 1000th iteration, the model has only seen approximately 16k images. This naturally raises an interesting question of whether *few-shot distillation* is feasible. In Appendix F, we explore this by training on a dataset of only 10 images (simultaneously using this as the batchsize), and find that the results remain comparable to those achieved using the full training set.

5. Nevertheless, there remains room for additional, less critical ablation studies—such as investigating the effect of the teacher-student mixing parameter $k$, which, as briefly mentioned in Section 4, serves to interpolate between trajectory-based and progressive-style distillation strategies. However, since the preceding ablations already demonstrate that accelerated self-distillation (i.e., non-zero $k$) leads to faster convergence, we consider this particular experiment to be of lower importance and do not explore it in detail.

We note that the above explorations primarily affect training speed, with negligible impact on the final converged results. Therefore, all the tables and figures presented in the experimental section remain valid and unaffected.

## 6 Conclusion and Future Work

In this work, we presented SCFM, an efficient distillation method that accelerates pre-trained flow matching diffusion models to very few steps. Experimental results demonstrate that our method best preserves many-step teacher characteristics. Compared to previous works, the computational and training data requirements is almost free, providing a compelling alternative to training-free acceleration methods while delivering significantly better results. Future directions may include:

- Preserving model creativity is a key aspect of distillation quality (i.e., producing diverse samples under different random seeds). Our experiments show that our method retains many-step teacher characteristics, including variability, though a quantitative metric is still lacking. This may stem from our focus on velocity trajectories, whereas common approaches directly predict clean samples (e.g., Song et al. [2023], Yin et al. [2024]) and yield less diversity. This also indicates that our method generalizes better in few-shot settings, being less affected by the scarcity of

training samples. Future work may combine both approaches to enhance sample quality and variability.

- While our approach eliminates the need for step-size embeddings in shortcut models with a distinct learning paradigm, it is nonetheless encouraging to see the development of highly efficient algorithms that converts pre-trained flow matching into exact shortcutting.
- Since ADD-style methods can be readily integrated into our framework to potentially enhance few-step generation, or even enable one-step generation, exploring a velocity-space variant of LADD may be a promising direction for enhancing velocity consistency.
- Our method is designed to be modality-agnostic and applicable to flow matching in diverse domains, such as 3D, video, audio and etc. It would therefore be valuable to explore how practitioners might adopt the algorithm, possibly with domain-specific modifications.

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

# A  Pseudo Code of Vanilla SCFM

First, let $L_n := \mathrm{Linspace}(1, 0, n + 1)$ denote the set of $n + 1$ equally spaced discretized timesteps, decreasing from 1 to 0, where $n$ corresponds to the target number of teacher steps.

Prevalent pre-trained flow matching models often avoid using uniformly spaced discrete ODE steps, instead applying timestep shifting to improve sample quality Esser et al. [2024], BlackForestLabs [2024]. In some cases, the shift may even dynamically depend on image resolution. In this paper, we focus exclusively on the non-dynamic shift, where the shifted timestep is defined as:

$$\mathrm{S}_s(t) = \frac{st}{1 + (s - 1)t}, \ s > 1. \tag{17}$$

For the shift parameter $s$ in (17), we sample it uniformly from the range $[2.5, 4.5]$ for both Flux and SD3.5-Large. Intuitively, a larger shift value $s$ concentrates more sampling steps in the high-noise region (early timesteps), and fewer in the data region (late timesteps), which typically leads to improved empirical performance compared to uniform scheduling.

While one could design a shift distribution tailored to the number of ODE steps, we observe that uniform sampling within this range performs comparably to more sophisticated strategies. Nonetheless, this remains an interesting direction for future exploration. The overall training procedure for the vanilla SCFM is summarized in Algorithm 1.

---

**Algorithm 1** Vanilla SCFM Training

---

1: **Input:** Teacher parameters $\boldsymbol{\theta}^*$, trainable student parameters $\boldsymbol{\theta}$, stopgrad parameters $\boldsymbol{\theta}^- \leftarrow \boldsymbol{\theta}$, discretized timesteps $L_n$, number of teacher-guided samples $k$, and batchsize $N$
2: **while** not converged **do**
3:     Sample data $\mathbf{x}_0 \sim \mathcal{D}$, and noise $\mathbf{x}_1 \sim \mathcal{N}(0, \mathbf{I})$
4:     Sample shift value $s \in [2.5, 4.5]$; $\forall t \in L_n$ apply shifting $t \leftarrow \mathrm{S}_s(t)$ as in (17)
5:     **for** first $k$ elements in the batch **do**
6:         Sample three consecutive $t_1, t_2, t_3$ from $L_n$ with a fixed skip of 1
7:     **for** remaining $N - k$ elements in the batch **do**
8:         Sample three consecutive $t_1, t_2, t_3$ from $L_n$ with a skip randomly drawn from $\{2, 4, \ldots, T/4\}$
9:     $\mathbf{x}_{t_1} \leftarrow (1 - t_1)\mathbf{x}_0 + t_1 \mathbf{x}_1$, then perform a forward pass to obtain $\mathcal{V}_{\boldsymbol{\theta}}(\mathbf{x}_{t_1}, t_1)$
10:     Compute targets $\mathcal{V}_{\boldsymbol{\theta}^*}(\mathbf{x}_{t_1}, t_1)$ and $\mathcal{V}_{\boldsymbol{\theta}^-}(\mathbf{x}_{t_1}, t_1)$ using (12), with either the embedded CFG or the variant in (16). Specifically, set $d_i \leftarrow t_1 - t_2$ and $d_{i+1} \leftarrow t_2 - t_3$.
11:     Backpropagate $\boldsymbol{\theta}$ using loss (13) with $\mathcal{V}_{\boldsymbol{\theta}}(\mathbf{x}_{t_1}, t_1)$, $\mathcal{V}_{\boldsymbol{\theta}^*}(\mathbf{x}_{t_1}, t_1)$ and $\mathcal{V}_{\boldsymbol{\theta}^-}(\mathbf{x}_{t_1}, t_1)$
12:     Update the EMA model $\boldsymbol{\theta}^-$ according to (14)
13:     Optional: If cyclic restarting is enabled, reinitialize $\boldsymbol{\theta}^- \leftarrow \boldsymbol{\theta}$ at fixed iteration intervals

---

# B  Closer Comparison to Other Trajectory-based Distillation Methods

As discussed in Section 2, shortcut distillation represents a novel class of trajectory-based distillation methods that operates in velocity space rather than directly regressing ODE-simulated samples. Existing trajectory distillation methods typically employ the following sample-space mappings:

$$f : (\mathbf{x}_t, t) \rightarrow \mathbf{x}_\tau, \tau \in [0, t), \tag{18}$$

which also requires ODE solving along trajectories. Unlike our velocity-space approach in (12), conventional methods Luo et al. [2023], Kim et al. [2024] enforce consistency through mixing teacher and student solvers:

$$\mathbf{x}_{t_{i+2}} = f_{\boldsymbol{\theta}^-}(\mathbf{x}_{t_{i+1}}, t_{i+1}) = f_{\boldsymbol{\theta}^-}\big(f_{\boldsymbol{\theta}^*}(\mathbf{x}_t, t_i), t_{i+1}\big). \tag{19}$$

Intuitively, this formulation injects teacher knowledge through the inner teacher ODE solver, and mixes it with the outer (stopgrad-)student solver. The student thus learns to generate outputs from arbitrary intermediate states produced by the teacher.

In contrast, shortcut distillation (9) employs consistent solvers for both steps—either using the teacher $\mathbf{x}_{t_{i+2}} = f_{\boldsymbol{\theta}^*}\big(f_{\boldsymbol{\theta}^*}(\mathbf{x}_t, t_i), t_{i+1}\big)$, or the student $\mathbf{x}_{t_{i+2}} = f_{\boldsymbol{\theta}^-}\big(f_{\boldsymbol{\theta}^-}(\mathbf{x}_t, t_i), t_{i+1}\big)$, and further combines

their outputs through self-distilled bootstrapping. This preserves internal solver consistency while progressively transferring knowledge from teacher to finer students. Finally, a generalized distillation loss can be written as

$$\mathcal{L}(\theta) = \mathbb{D}\big(f_{\boldsymbol{\theta}}(\mathbf{x}_{t_i}, t_i), f_{\boldsymbol{\theta}^-}(\mathbf{x}_{t_{i+1}}, t_{i+1})\big). \tag{20}$$

In light of this loss formulation, we proceed to elaborate on the key differences in greater detail.

**Remark 2.** For flow matching models, the ODE in (18) simplifies to (5). Henceforth, some previous works may arrive at objectives resembling $\mathbb{D}\big((d_i + d_{i+1})\mathcal{V}_{\boldsymbol{\theta}}(\mathbf{x}_{t_i}, t_i), d_i\mathcal{V}_{\boldsymbol{\theta}^*}(\mathbf{x}_{t_i}, t_i) + d_{i+1}\mathcal{V}_{\boldsymbol{\theta}^-}(\mathbf{x}_{t_{i+1}}, t_{i+1})\big)$, which superficially resembles our (12). However, crucial differences exist:

- $\mathbb{D}$ in existing works is measured in sample space rather than velocity space. Unless a sample-wise/adaptive weighting function proportional to $1/(d_i + d_{i+1})$ is explicitly incorporated, this formulation fails to align with the scaling behavior inherent to flow matching objectives. To our knowledge, no prior work has proposed such a weighting scheme specifically tailored to flow matching models, nor is there clear empirical evidence supporting its use.

- Numerically, since $d_i \in (0, 1]$ in (10), errors computed in sample space are typically 10–100× smaller than those in velocity space. As a result, they require extremely small learning rates (e.g., $10^{-6}$) and significantly more training iterations. For example, [Ren et al., 2024, Section 4.1] reports that each distillation stage for a 2.6B-parameter SDXL requires approximately 200 A100 GPU hours. Some works also consider alternatives to the standard $\ell_2$ loss for $\mathcal{D}$, such as the Huber loss, $\ell_1$ loss, or perceptual losses like LPIPS Zhang et al. [2018]. This is usually motivated by the observation that sample-space losses often exhibit greater instability. As a result, more robust loss functions—such as the Huber loss—are sometimes preferred due to their reduced sensitivity to outliers (e.g., see Chen et al. [2025]). In our case, we adopt the standard $\ell_2$ loss, which preserves the optimality of unbiased estimation in velocity fields, as hinted in Section 3.1.

Nonetheless, as shown in Appendix C, we explore the possibility of mixing only the teacher solver with the student—following the practice of some prior works—and observe nearly identical results with slightly faster convergence. This finding motivates a deeper investigation into how to best leverage self-distillation, as further explored in Appendices D and E.

## C   Ablation on The First Term of Loss (13)

As hinted in Appendix B and in the first point of Section 5.4, we explore an alternative formulation in which the solvers used in the first term of (13) are interleaved or mixed, as described in (19), while keeping the second term unchanged. More formally, similar to Equation (12), the modified target $\mathcal{V}_{\boldsymbol{\theta}^*}(\mathbf{x}_{t_i}, t_i)$ is computed as

$$\mathcal{V}_{\boldsymbol{\theta}^*}(\mathbf{x}_{t_i}, t_i) = \frac{d_i}{(d_i + d_{i+1})}\mathcal{V}_{\boldsymbol{\theta}^*}(\mathbf{x}_{t_i}, t_i) + \frac{d_{i+1}}{(d_i + d_{i+1})}\mathcal{V}_{\boldsymbol{\theta}^-}(\mathbf{x}_{t_{i+1}}, t_{i+1}). \tag{21}$$

This formulation linearly blends teacher and EMA student together, enabling faster knowledge transition to the second term of $\mathcal{L}_{\text{scfm}}$. As an evident in Figure 3, this approach (denoted as vanilla-mix) yields slightly faster convergence compared to the vanilla SCFM formulation. This observation suggests that employing a more frequently updated EMA student model may facilitate better knowledge propagation from the teacher.

To further investigate this idea, we introduce a cyclic restarting strategy for the student EMA model. In this scheme, the EMA accumulator is periodically reset at fixed iteration intervals, allowing the student to forget stable or overly smoothed knowledge and adapt more quickly to the evolving teacher signal. This leads to even faster convergence, as discussed in detail in Appendix D.

## D   Ablation on Cyclic Restating

Cyclic restarting is included as an optional component in Algorithm 1 (highlighted in blue), designed to accelerate training by periodically resetting the EMA student model. In this section, we present experimental evidence demonstrating the effectiveness of this simple yet practical strategy. Specifically, we compare the vanilla version of Algorithm 1 against two variants: one with an

---
**Algorithm 2** Fast-Slow EMA SCFM Training

---
1: **Input:** Teacher parameters $\boldsymbol{\theta}^*$, trainable student parameters $\boldsymbol{\theta}$, fast stopgrad parameters $\boldsymbol{\theta}^+ \leftarrow \boldsymbol{\theta}$ with $\mu = 0.99$, slow stopgrad parameters $\boldsymbol{\theta}^- \leftarrow \boldsymbol{\theta}$ with $\mu = 0.999$, discretized timesteps $L_n$, number of teacher-guided samples $k$, and batchsize $N$
2: **while** not converged **do**
3:     Sample data $\mathbf{x}_0 \sim \mathcal{D}$, and noise $\mathbf{x}_1 \sim \mathcal{N}(0, \mathbf{I})$
4:     Sample shift value $s \in [2.5, 4.5]$; $\forall t \in L_n$ apply shifting $t \leftarrow \mathrm{S}_s(t)$ as in (17)
5:     **for** first $k$ elements in the batch **do**
6:         Sample three consecutive $t_1, t_2, t_3$ from $L_n$ with a fixed skip of 1
7:     **for** remaining $N - k$ elements in the batch **do**
8:         Sample three consecutive $t_1, t_2, t_3$ from $L_n$ with a skip randomly drawn from $\{2, 4, \ldots, T/4\}$
9:     $\mathbf{x}_{t_1} \leftarrow (1 - t_1)\mathbf{x}_0 + t_1\mathbf{x}_1$, then perform a forward pass to obtain $\mathcal{V}_{\boldsymbol{\theta}}(\mathbf{x}_{t_1}, t_1)$
10:    Compute the targets $\mathcal{V}_{\boldsymbol{\theta}^*}(\mathbf{x}_{t_1}, t_1)$ and $\mathcal{V}_{\boldsymbol{\theta}^-}(\mathbf{x}_{t_2}, t_2)$ using (21) and (22). Remaining calculations follow the same as Line 10 of Algorithm 1
11:    Backpropagate $\boldsymbol{\theta}$ using loss (13) with $\mathcal{V}_{\boldsymbol{\theta}}(\mathbf{x}_{t_1}, t_1)$, $\mathcal{V}_{\boldsymbol{\theta}^*}(\mathbf{x}_{t_1}, t_1)$ and $\mathcal{V}_{\boldsymbol{\theta}^-}(\mathbf{x}_{t_1}, t_1)$
12:    Update the EMA models $\boldsymbol{\theta}^-$ and $\boldsymbol{\theta}^+$ according to (14), each using its respective decay rate

---

aggressive restarting schedule that resets the EMA every 500 iterations (referred to as cyc-500), and another with a more moderate schedule that resets every 2000 iterations (denoted as cyc-2000).

As shown in Figure 3, both variants significantly improve convergence speed over the vanilla baseline. The cyc-500 variant exhibits the fastest initial convergence due to its rapid adaptation, while cyc-2000 achieves a more balanced trade-off between stability and speed. These results confirm that periodic forgetting of stale EMA knowledge can facilitate quicker adaptation and reduce training time.

However, as discussed in Appendix D and illustrated in Figure 4, overly frequent resets (e.g., cyc-500) can lead to performance degradation by hindering late-stage convergence. This highlights the need for careful tuning of the restart interval, which may vary depending on the specific distillation setup. To address this issue, and inspired by the findings in Appendix C, we propose a fully automatic fast training algorithm, presented in Appendix E.

## E  Final Version: Ablation on Dual Fast-slow EMA

We retain the computation of $\mathcal{V}_{\boldsymbol{\theta}^*}(\mathbf{x}_{t_i}, t_i)$ as defined in (21), but introduce a fast EMA model, denoted by $\boldsymbol{\theta}^+$, into the second term of the SCFM loss. This modification still preserves the solver consistency in terms of self-distillation, with the key distinction being that we now mix outputs from both a faster and a slower EMA model. Specifically, the revised computation of the fast-slow target $\mathcal{V}_{\boldsymbol{\theta}^-}$ is given by

$$\mathcal{V}_{\boldsymbol{\theta}^-}\big(\mathbf{x}_{t_i}, t_i\big) = \frac{d_i}{(d_i + d_{i+1})}\mathcal{V}_{\boldsymbol{\theta}^+}\big(\mathbf{x}_{t_i}, t_i\big) + \frac{d_{i+1}}{(d_i + d_{i+1})}\mathcal{V}_{\boldsymbol{\theta}^-}\big(\mathbf{x}_{t_{i+1}}, t_{i+1}\big), \tag{22}$$

The resulting dual stopgrad EMA training algorithm largely mirrors the structure of Algorithm 1; hence, we highlight only the modified components in red for clarity and ease of comparison.

As shown in Figure 3, Algorithm 2 achieves even faster convergence—outperforming the aggressively restarted variant of Algorithm 1 that resets every 500 training iterations. Most importantly, an additional benefit is that this fast training procedure is fully automatic and avoids the late-stage performance degradation seen with the manual restarting strategy, as demonstrated in Figure 4.

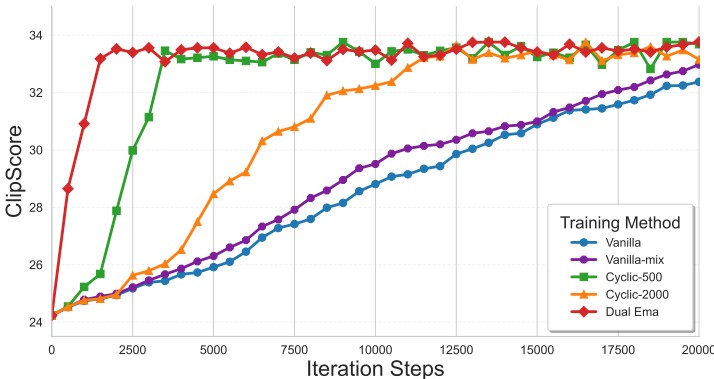

Figure 3: CLIP score progression across training methods, evaluated on 20 random image-text pairs from the COCO-30k validation set. Experiments are conducted on Flux and evaluated on the 8-step student.

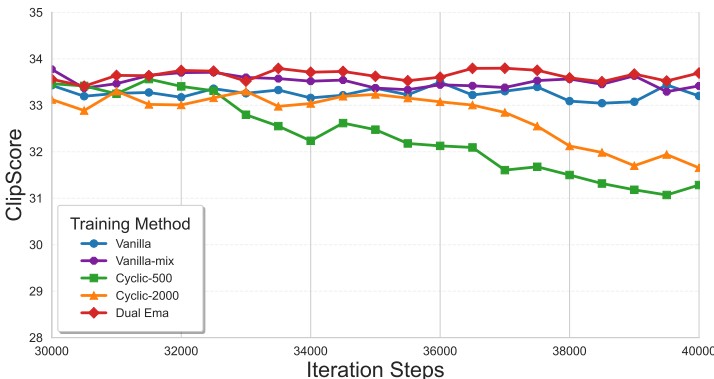

Figure 4: Same setting to Figure 3, focuses exclusively on the convergence behavior during the later stages of training.

# F    Ablation on Few-shot Learning

In this section, we present an interesting ablation study to evaluate the effectiveness of SCFM in a few-shot learning setting. As shown in Table 2, distilling an 8-step student model using a dataset of only 10 images results in only minor differences in CLIP score, along with a slight increase in FID and $\Delta$FID compared to the baseline trained on the full dataset. However, these quantitative differences do not fully capture the performance of the model. As illustrated in Figure 5, the few-shot distilled model demonstrates only modest degradation in *preservation ability*. In the mean time, the generated samples maintain high image quality and strong prompt alignment, aligning well with the metrics in Table 2. It is also worth noting that preservation ability may not always be a critical goal—especially in cases where the objective is to guide the student model to learn from a specific, small dataset, or to acquire general generation capability rather than closely imitating the teacher model.

Overall, this suggests that while the shortcutting ability of SCFM may be moderately constrained when training data lacks sufficient coverage of the teacher's knowledge, the overall performance remains surprisingly robust. Such a setting could be particularly valuable when collecting large-scale data is impractical, e.g., 3D, video or even robot data. In such cases, even a limited set of self-generated or synthetic samples might be sufficient to produce a strong distilled model. A more thorough exploration of these data-efficient regimes is left for future investigation.

Table 2: Qualitative evaluation of few-shot SCFM trained on Flux. All samples are generated using 8 steps. The experimental setup follows the same configuration as in Table 1.

|  | ΔFID↓ | FID↓ | CLIP↑ |
|---|---|---|---|
| Flux.1-Dev | 27.43 | N/A | 33.60 |
| SCFM | **+0.16** (27.59) | **2.58** | **33.76** |
| **Few-shot** SCFM | -0.3 (27.13) | 2.94 | 33.75 |

| **Flux-Dev** | **SCFM** | **Few-shot SCFM** |
|---|---|---|

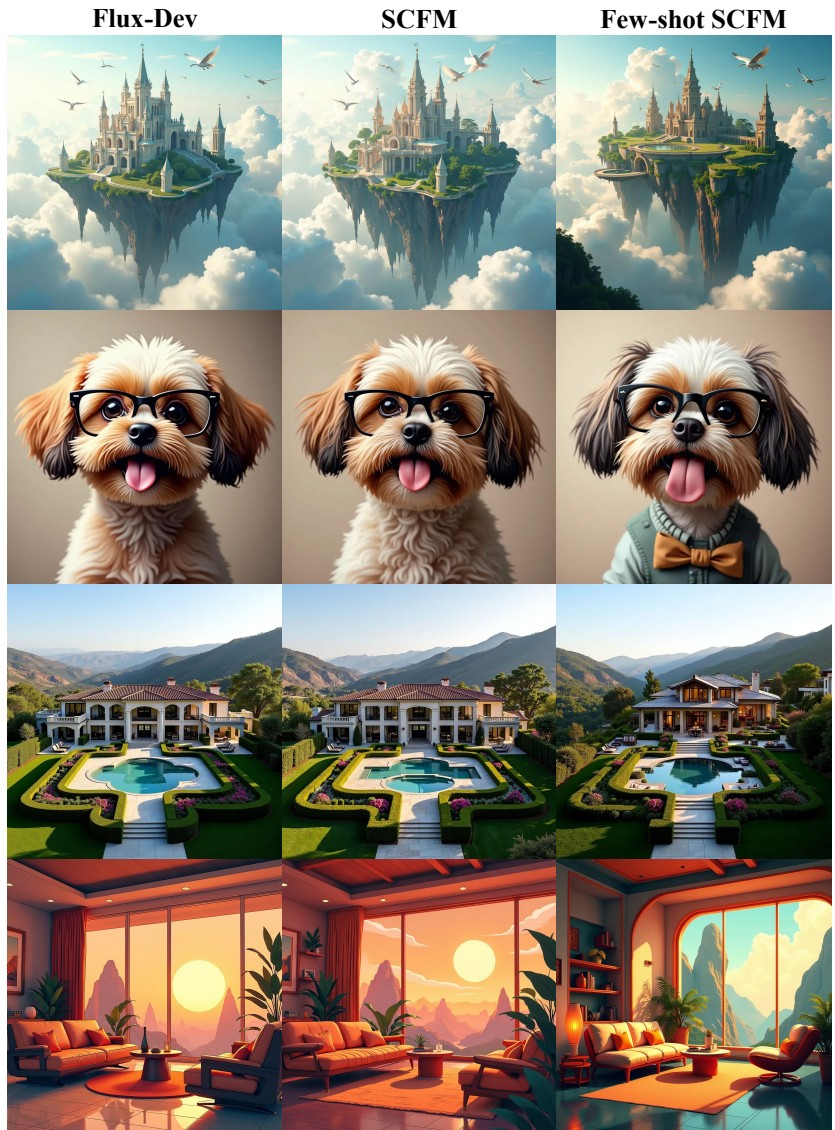

Figure 5: Side-by-side comparison on few-shot trained v.s. regularly trained SCFM. Teacher samples use 32 steps, whereas others use 8.

## G   Comments on CLIP Score

In this section, we elaborate on our choice to use the CLIP checkpoints provided by Koukounas et al. [2024]-named Jina-rather than the official OpenAI versions. Although OpenAI's CLIP models are widely adopted, we found that their scoring often fails to align with human perceptual or semantic

judgments—particularly in the case of low-quality synthetic data. While our validation set contains numerous examples, a representative case in Figure 6 demonstrates that the Jina checkpoint is the only variant among those tested that produces a CLIP score ranking consistent with human preference. In contrast, the OpenAI models fail to capture key elements from the caption, such as "grass", mistakenly favoring an image that shows green land but no actual grass. This suggests that the model may be overly influenced by coarse visual features. Such discrepancies indicate that the Jina checkpoint provides better calibration for text-image similarity, making it a more appropriate choice for our evaluations. Nevertheless, all CLIP scores obtained in Table 1 uses this model consistently.

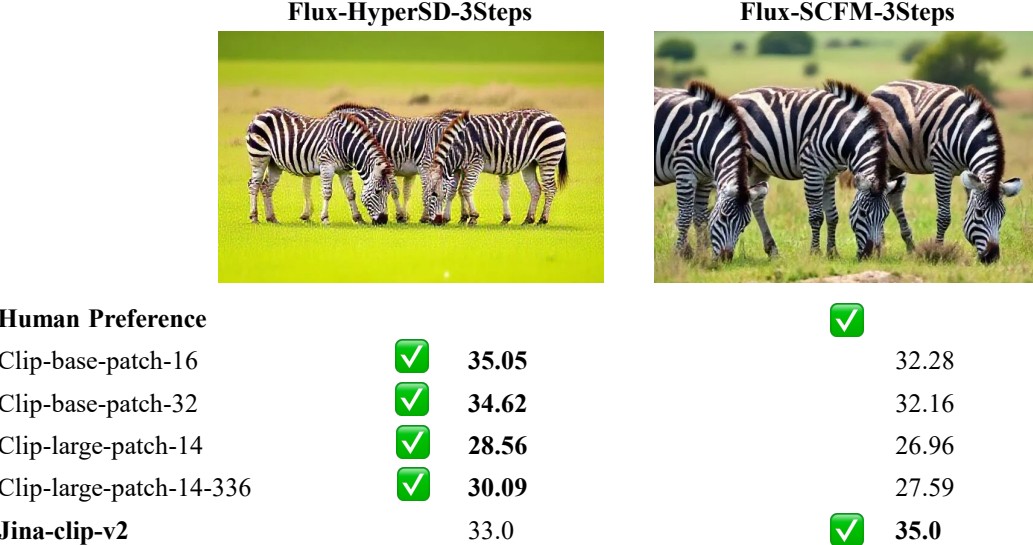

| | Flux-HyperSD-3Steps | Flux-SCFM-3Steps |
|---|---|---|
| **Human Preference** | | ✅ |
| Clip-base-patch-16 | ✅ **35.05** | 32.28 |
| Clip-base-patch-32 | ✅ **34.62** | 32.16 |
| Clip-large-patch-14 | ✅ **28.56** | 26.96 |
| Clip-large-patch-14-336 | ✅ **30.09** | 27.59 |
| **Jina-clip-v2** | 33.0 | ✅ **35.0** |

Figure 6: The caption used to generate the two images above is: "A group of zebras grazing in the **grass**."

# H   Additional Results for Flux

See the visual comparisons in Figures 7, 8, and 9, where we only include the TeaCache-0.8 results in Figure 7, as it offers inference speed closest to that of 8-step distilled models (see Table 1).

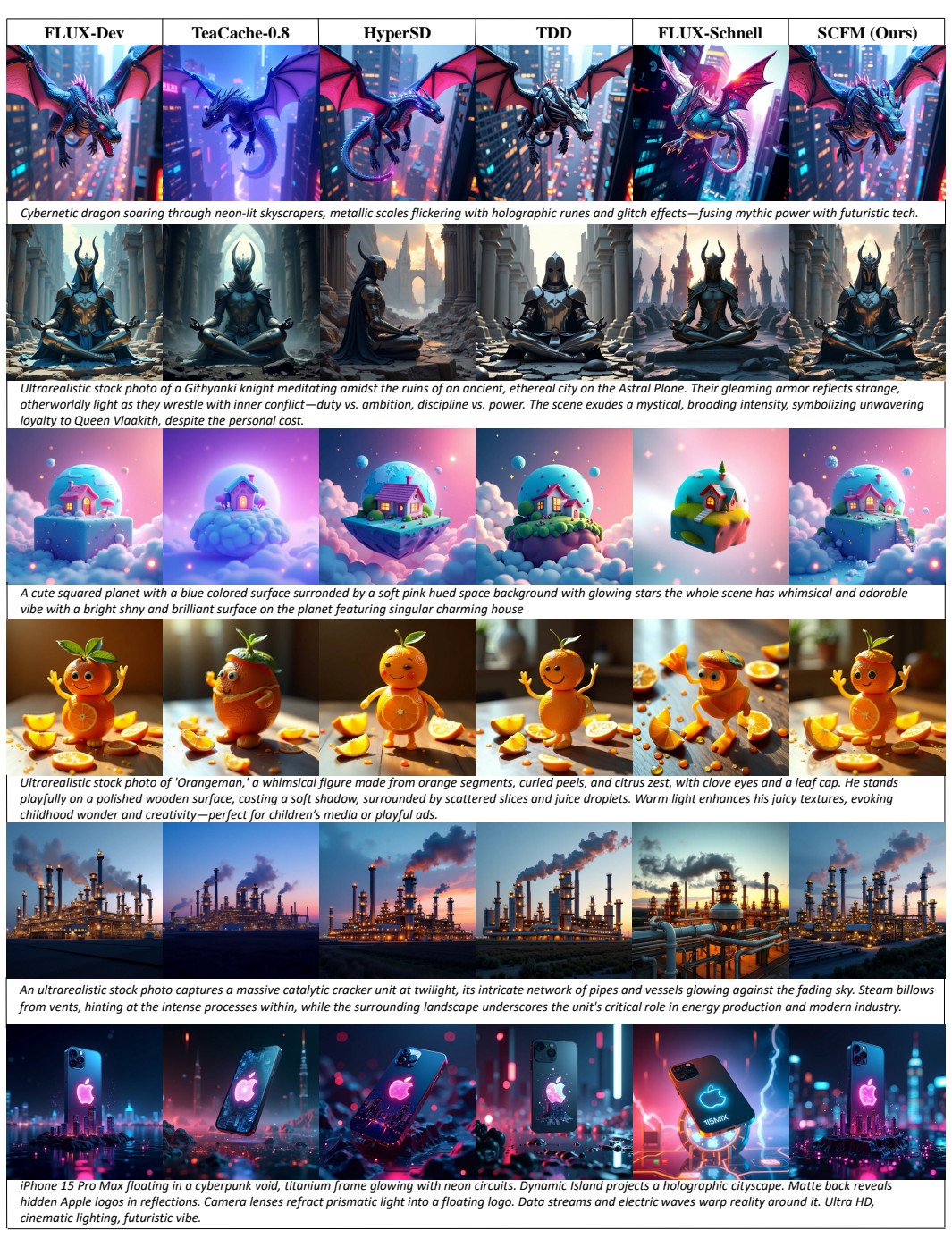

Figure 7: Additional visual comparisons for Flux under 8 steps.

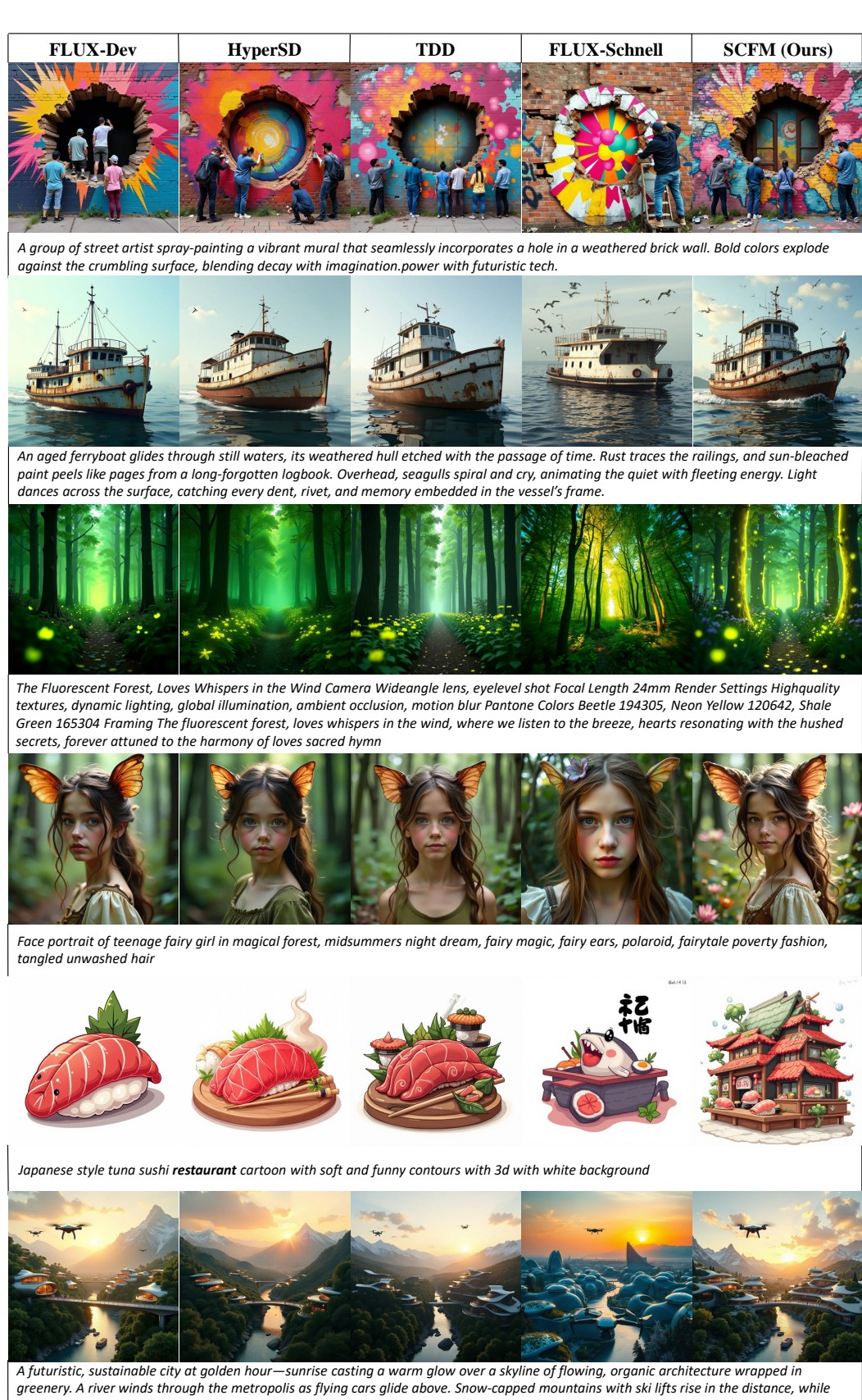

Figure 8: Additional visual comparisons for Flux under 4 steps.

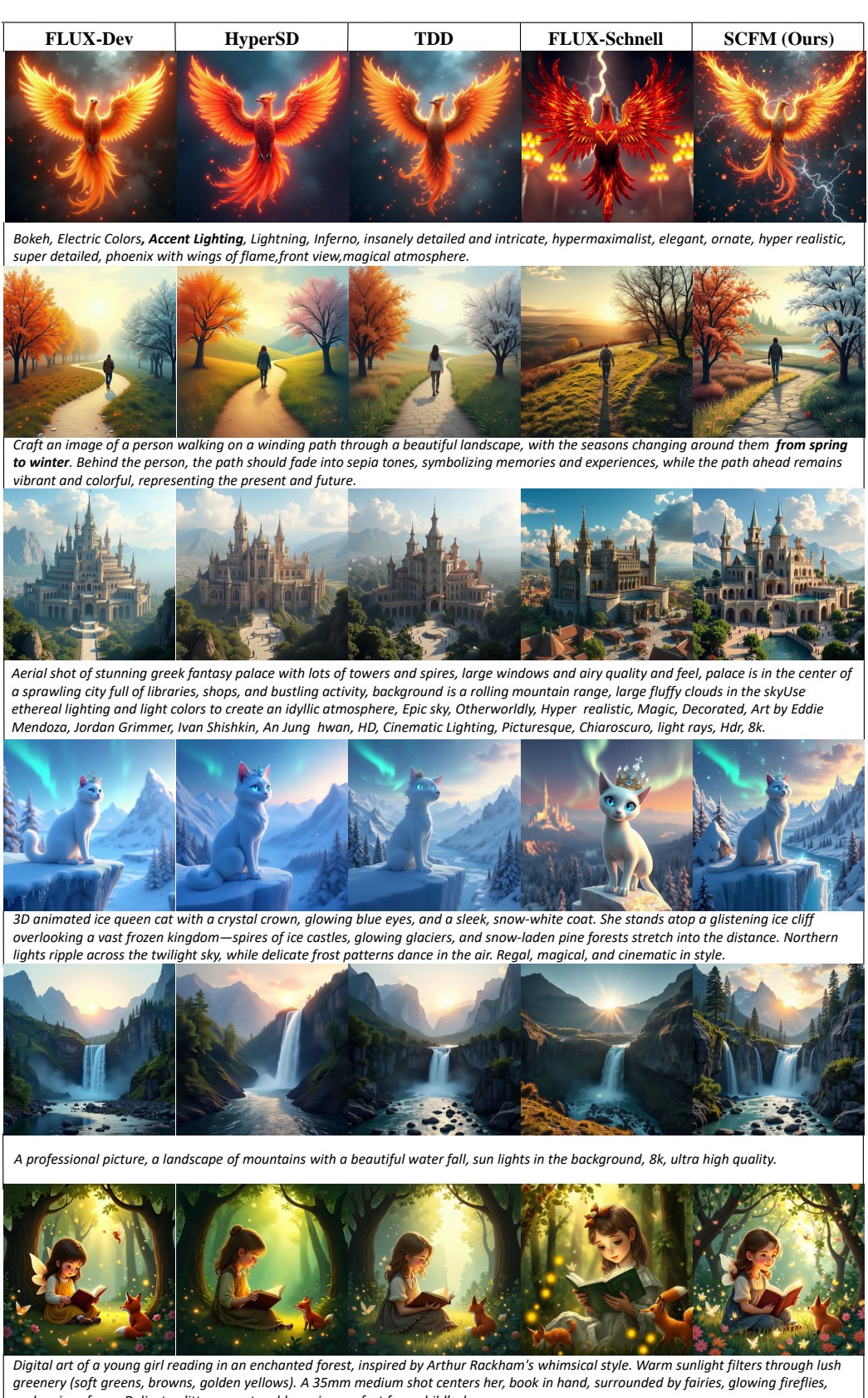

Figure 9: Additional visual comparisons for Flux under 3 steps.

# I Additional Results for SD3.5L

See the visual comparisons in Figure 10.

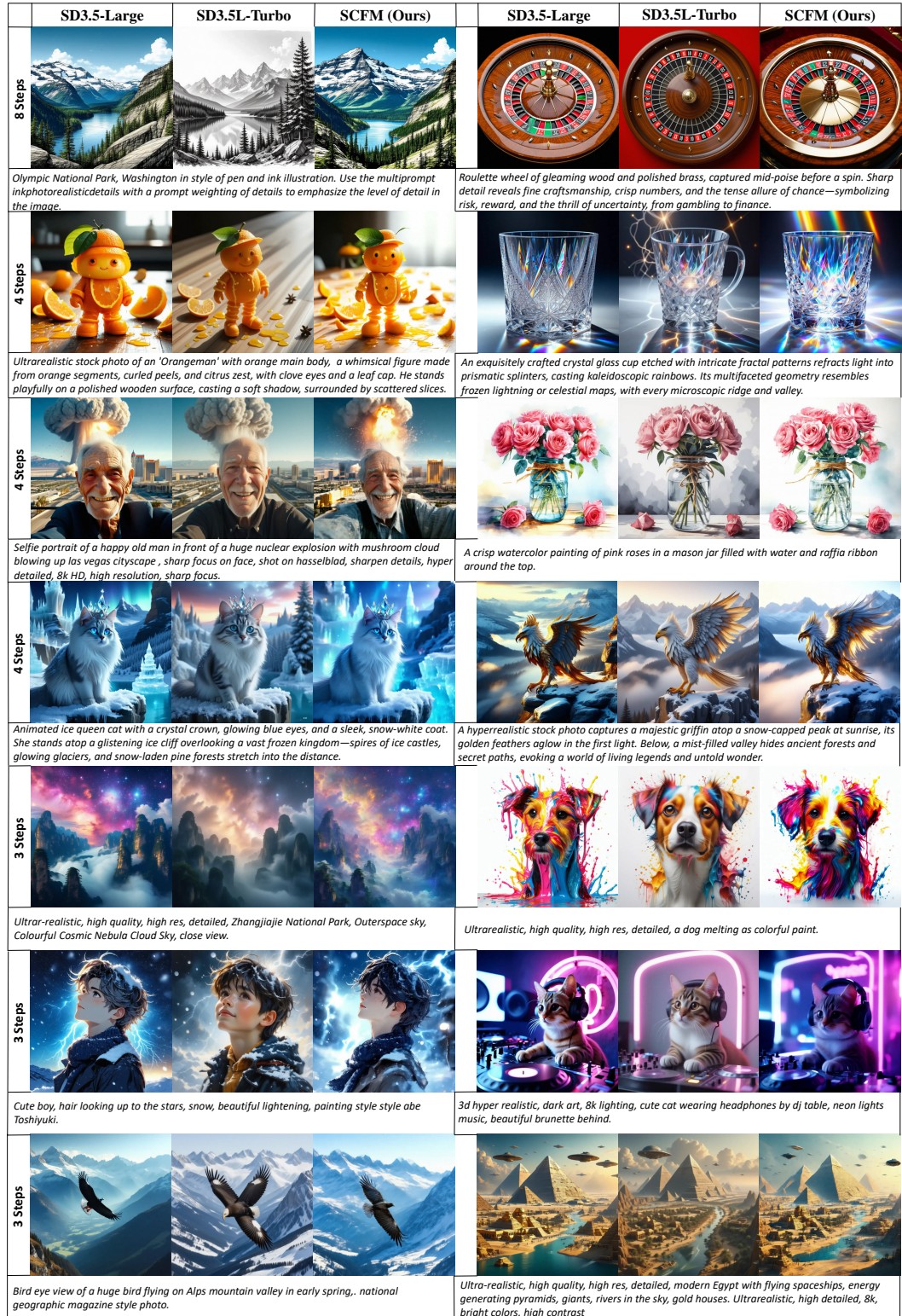

Figure 10: Additional visual comparisons for SD3.5-Large.

