# OpenReview forum: "Shortcutting Pre-trained Flow Matching Diffusion Models is Almost Free Lunch"
_NeurIPS.cc/2025/Conference — NeurIPS 2025 poster_

### Official Review · Reviewer_SAAF · 2025-06-29

**Clarity:** 3
**Significance:** 3
**Originality:** 3
**Rating:** 4
**Confidence:** 4

**Summary:**

The paper proposes a method to conduct short-cut distillation on pre-trained Flow Matching based diffusion models. The goal is to get rid of the explicit variable which causes the original model structure to be modified. The method is done through a weighted sum of flow matching loss and a self-consistency loss. Experiments show that the proposed method achieve best performance compared to baseline methods and maximally achieves similar output to the ones generated by the teacher model.

**Questions:**

1. How does the proposed method work under extremely low time steps such as 1
2. What's the author's plan for releasing the code and checkpoints for further evaluation?
3. For the rest of my questions, please see the weakness part.

**Ethical Concerns:**

["NO or VERY MINOR ethics concerns only"]

**Final Justification:**

The paper proposed a way to accelerate model inference by shortcutting the model which doesn't require extensive re-training. The achieved performances are strong. Therefore I maintain my positive rating and kindly ask the authors to release their code as promised.

**Limitations:**

yes

**Quality:**

3

**Strengths And Weaknesses:**

## Strength
1. The paper is well written and easy to follow.
2. The proposed method is simple and effective with low training cost
3. The final performance is strong even without ADD/LADD
4. The presented method requires no changes to the original model

## Weakness
1. There is no comparison with the original short-cut model.
2. Some ablation studies are only briefly discussed without conducting the actual experiment such as integrating with ADD or LADD.
3. The author mentioned that the proposed method is not limited to image modality but no experiments are conducted to show the effectiveness of the proposed method in other modalities such as video.
4. The evaluated models (SD-3.5 Large 8B and Flux.1 Dev 12B) are limited, it will be good show that the proposed methods work for other open-source models too such as smaller models and compare with more distillation methods

---

> ### Author Rebuttal · Authors · 2025-07-29
>
> We are grateful to have received positive feedback.  Some of our responses overlap with those addressed to other reviewers, and we provide cross-references for clarity and conciseness.  Please feel free to raise your score if you find that your concerns have been addressed or if you see new strengths, interesting insights, or future potential in our work :) .
>
> Q1.*Compare to original short-cut model*
> A1. We attempted to adapt the original shortcut approach to our distillation setting, but it yielded poor convergence and low-quality outputs.  For details, we kindly refer the reviewer to our response to Question 1 from Reviewer qe9Q for a detailed explanation.
>
>
> Q2. *Lower steps / ADD*
> A2. Please see our response to Reviewer VfPo, Question 2, where we address the reasons in detail.
>
> Q3. *Other baseline models/ distillation methods*
> A3. A detailed discussion of this can be found in our response to Question 3 from Reviewer VfPo.
>
> Q4. *Code release*
> A4. We commit to releasing the code following the final decision on the paper. We would also appreciate some time to further develop and refine it to ensure it is released in its best possible form. Please also see our response below for more information.
>
> Q5. *Experiment with other modalities*
> A5. Related to the code releasing question, we are planning to release a general-purpose training library based on our algorithm. This library is designed to support a wide range of customized modalities and models, rather than implementing each modality separately, which we find inefficient. To our knowledge, this would be the first off-the-shelf codebase that supports such flexibility. For example, in the case of video, our method requires only a few self-generated clips. It also supports distillation for modality-conditioned variants, such as the recently released FLUX-Kontext.

---

> > ### Comment · Reviewer_SAAF · 2025-08-03
> >
> > Thank you for the response. I still think the paper needs more experiments to justify its performance especially without releasing its source code which is a concern shared by other reviewers too, but the current approach provides a promising direction that's worth showing to the public.

---

> > > ### Author Response · Authors · 2025-08-05
> > >
> > > Thank you for your thoughtful feedback. We appreciate your recognition of the potential in our approach. We are actively working on preparing the open-source repository and commit to releasing the full source code and instructions upon acceptance to ensure reproducibility. As per the rebuttal policy, we are not permitted to include any links or make updates to the submitted GitHub repository during this period.

---

### Official Review · Reviewer_VfPo · 2025-07-02

**Clarity:** 3
**Significance:** 3
**Originality:** 3
**Rating:** 4
**Confidence:** 3

**Summary:**

This paper integrates the self-consistency loss from shortcut flow matching ideas into model distillation in the velocity space of flow matching models. The authors additionally uses an implicit step size approach to improve training efficiency. The method is evaluated on certain latest text-to-image models.

**Questions:**

1. See the "Weakness" above.

2. Does the training of the proposed model always initialize the model weights from the pre-trained model? Are prompts used in the test time different from the training time? Especially, when you talk about training with only 10 images and achieving good results, do you also use the same 10 prompts/images for testing or do you use different prompts/images?

3. What is the model performance on versions of Stable Diffusion (e.g. 1.5) where there are already established baselines?

4. Why does the number of steps stop at 3? Why not lower?

**Ethical Concerns:**

["NO or VERY MINOR ethics concerns only"]

**Final Justification:**

Thanks for the responses. Nevertheless, I think that training a single $V_\theta(x_t, t)$ across all $t$ will theoretically require straight trajectories, while training different $V_\theta(x_t, t)$ parameterized by $t$ only requires local straightness. I will maintain my rating.

**Limitations:**

Yes

**Quality:**

3

**Strengths And Weaknesses:**

**Strengths:**

-- Distillation a large pre-trained flow matching diffusion model into a model with fewer steps is an important research direction.

-- Incorporating the self-consistency loss from shortcutting flow matching ideas into distillation is nice and makes a lot of sense.

-- The method is evaluated on certain latest text-to-image models.

**Weaknesses:**

-- The proposed model integrates the self-consistency idea from shortcut flow matching approaches but with an implicit step size. In particular, previous shortcut flow matching model trains for the look-ahead velocity V(x_t, t, d) where d is the step size. This does not align with a pretrained model. This paper instead proposes to train for a single velocity of the form v_\theta(x_t, t) for all step sizes (what they refer to as implicit step size $d$, based on my understanding). I find this somewhat problematic: In general, this is ill-defined, because if the trajectory is not straight, then V_\theta(x_t, t, d) may vary with different $d$; that is, there does not exist a single V_\theta(x_t, t) for all different step size. The paper did try to motivate their method by assuming that we can straighten curved velocity trajectories. However, I am not sure how justified this is, especially since the teacher model in the first term of Eqn(13) may not have straight trajectories. Furthermore, as this paper states in lines 86-87: "In practice, however, class-conditional models often deviate from this idealized trajectory, especially in high-noise regions where the mapping often exhibit curvature".

-- I feel that the proposed model can be more comprehensively evaluated, such as comparing with popular distillation methods, e.g. reflow, progressive distillation, consistency distillation, on the pre-trained models where there are already established baselines. I would imagine that this should not be hard given the proposed method is reasonably efficient.

-- In Table 1, the $\Delta FID$ of the newly proposed SCFM often is not the best for Flux related models. For SD3.5 model, the SCFM doesn't see to have significant improvement for low step sizes, but with a higher latency.

---

> ### Author Rebuttal · Authors · 2025-07-29
>
> We thank the reviewer for the positive feedback and these constructive questions, which are valuable for clarifying key aspects of our work.  We hope that our responses help clarify your questions and showcase the potential of our work. If so, we kindly invite you to consider raising your score :).
>
> Q1. *$V_\theta(x_t, t)$ can be ill-defined*
> A.1 To clarify, training a single velocity model $V_\theta(x, t)$ across multiple step sizes (i.e., without explicit $d$-conditioning) is well-defined within the framework of rectified flow training.  Specifically,  the 1-rectified flow model is derived from a family of $k>1$-rectified flows, each inducing **different** (and possibly curved) velocity fields.  While each $k \ge1 $ version has its own teacher and potentially distinct $V_\theta$, in practice, one may recover the 1-rectified version either by (1) gradually updating the model as $k$ decreases, or (2) training a unified model across all $k \ge 1$ trajectories (**as we do**). These two approaches, when fully optimized, should theoretically yield similar outcomes, and justify each other.
>
> Given this view, we believe it is more precise to reframe the concern: rather than “assuming we can straighten curved velocity,” our method encourages the model to learn to straighten trajectories from curvature.  In this sense, straightness is not assumed: it emerges as a learned property from training using our proposed training objective. This is achieved by rectifying curved trajectories as much as possible. Indeed, in worst-case scenarios, e.g. regions with extreme curvature, the training loss can become large and may require special treatment.  In these cases, as the reviewer pointed out, the model may struggle to align all the $V_\theta(x, t)$ consistently. A more comprehensive justification of such behavior likely requires further theoretical investigation in future work.
>
>
> Q2. *Why not lower steps*
> A2. Based on our understanding on existing low-step generation methods, achieving high-quality images with extremely low sampling steps requires two key components: **core algorithm** and **image-level feature learning**.  Current one/two-step generation methods rely heavily on optimizing feature learning through additional modules such Adversarial Distillation (ADD), which injects richer discriminative visual features into the model. This reflects the fact that reducing sampling steps is not solely a matter of core algorithm design (**which is our focus**), but also requires enhanced perceptual understanding.
>
>
> For this reason, all competing baselines focus on ADD to overcome lower step challenges , and may sometimes step in their core algorithm contribution (*something we aim to avoid in our approach*).  Nevertheless, we hypothesize that there may exist more elegant alternatives to replace ADD by focusing on more straightforward image feature learning. We are actively exploring this direction and hope to report promising results in the near future.
>
> Additionally, we envision SCFM as a lightweight yet vital *post-training refinement* procedure (analogous in spirit to RLHF), that can significantly reduce sampling steps while preserving or even improving generative quality.
>
>
> Q3. *Not comprehensively evaluated; Other baseline models/ distillation methods*
> A3. Thank you for the suggestion. There are several reasons why we did not include additional experiments with methods such as ReFlow, Progressive Distillation (PD), and Consistency Distillation (CD):
> - These distillation techniques have so far only been implemented on EDM-based models, while our method is specifically designed for flow-matching approaches. As a result, a direct and fair comparison is not feasible. The same reasoning applies to SD1.5, which is not pretrained using flow matching and therefore not directly comparable to our setting.
> - We believe the most meaningful comparison would require a benchmark consisting of toy-level pretrained flow-matching models that could then be distilled using PD, CD, and other approaches. Creating such a benchmark would likely need broader community support to establish a shared set of FM-pretrained models.
> - Our work focuses on large-scale pretrained flow-matching models. Implementing and evaluating all baselines at this scale would require computational resources that we currently cannot afford.
> - We note that these distillation techniques have already been shown to perform worse than the baselines listed in Table 1 in their original papers. It is therefore reasonable to expect that our method would also outperform them under similar conditions.
>
>
> Q4. *Some results are not the best*
> A4. There may be some misunderstanding in our presentation, but as shown in Table 1, our method consistently *outperforms* all baselines across the reported metrics. The doubled latency in the SD3.5 experiments, as explained in lines 222–226, results from skipping the CFG distillation (unlike SD3.5L-Turbo which includes it), and consequently performing inference with double batch sizes.  We will revise the phrasing to clarify this distinction and avoid any further confusion.
>
>
> Q5. *Model weights initialization; training/testing prompt difference*
> A5. The model is trained using LoRA, with the $B$ matrix initialized to zero. Therefore, in all our experiments, the model weights are indeed initialized from the pretrained backbone.
>
> Regarding training data, the training and testing prompts are **different**, in other words, the model is evaluated in a fully **blind** setting. The few-shot learning experiments are intended to show that a **general distillation capability** can be obtained using a small, randomly selected dataset, or even one generated by the model itself. The benefits are obvious: (1) unlike all other methods, we do not require big training sets especially for uncommon modalities; (2) less computational resources; (3) appealing flexibility for model expansion (see our response to Reviewer SAAF for more details); (4) customizable to any self-maintained/private data requirements, and etc.

---

> > ### Comment · Reviewer_VfPo · 2025-08-03
> >
> > Thank you for the response. I have no further questions.

---

### Official Review · Reviewer_qe9Q · 2025-07-03

**Clarity:** 2
**Significance:** 3
**Originality:** 3
**Rating:** 5
**Confidence:** 3

**Summary:**

The authors consider shortcut models and note that while they can be used to obtain a few-step flow model, they require training from scratch, since they require a specialized step-size embedding incompatible with existing diffusion/flow models. To address this problem, the authors propose a novel distillation objective that can be used for “shortcutting” a model without using additional “step size” embedding and thus is applicable for fine-tuning of already pretrained large text-to-image models. The authors evaluate their method together with LoRA decomposition of parameters to distill SD3.5L and Flux models.

**Questions:**

No

**Ethical Concerns:**

["NO or VERY MINOR ethics concerns only"]

**Final Justification:**

I keep my original score. During the rebuttal, the authors answered my questions.

**Limitations:**

Yes

**Quality:**

3

**Strengths And Weaknesses:**

**Strengths:**

The authors propose to extend the “shortcutting” procedure to the use of pretrained models, which do not have
The authors support the proposed method by testing it on distilling SD3.5L and Flux models.
The proposed method (together with using LoRA) is efficient and allows for efficient distillation of large models like Flux into a few-step generator under 24 A100 GPU hours.

**Weaknesses:**
- The main benefit of the proposed algorithm is that it is applicable to the original model without architectural modifications. However, it is not clear whether a direct addition of the step-size conditioning with randomly initialized parameters to the pretrained teacher model (+ using EMA for teacher, as in the proposed method) will be less efficient or provide worse quality of generation. Furthermore, the direct addition of the embedding conditioning does not require
- Notation consistency. The authors start to derive the self-consistency principle using notation V_{\theta} and then change it to V_{\theta_{-}} (Eq. 11 and Eq. 12). I think it would be better to use a general notation for the velocity in this derivation and introduce parametrization of teacher/EMA later, when discussing the algorithm to make notation consistent.
- Notation of sums in Eq.13 is ambiguous. There are two sums, but it is hard to understand the summation of what they describe, since there is no index of summation in the sums and in the expressions of these sums. In essence, $N$ is a batch size, and $k$ is the number of elements from the batch size used for the first part, and $N-k$ is the number of elements used for the second term. It should be written on the top of sums as their limit, and also a summation index should be introduced (e.g. $\sum_{n=1}^{k}$, $\sum_{n=k+1}^{N}$ and $n$ is used in the loss expression). Since it is the main expression of the paper, it is critical to provide it in the most clear form.

---

> ### Author Rebuttal · Authors · 2025-07-29
>
> We are very grateful to the reviewer for the feedback, and very pleased to have received positive reviews.
>
> Q1. *A direct shortcut implementation*
> A1. We have attempted to implement the original shortcut method for comparison. However, we observed that convergence was significantly slower and the resulting performance was not competitive with any other baselines (i.e., it failed to produce reasonable outputs at low sampling steps, such as 8-12).  Henthforce, applying the original shortcut formulation to large models such as FLUX or SD3.5 appears to require substantial computational resources and/or non-trivial pretraining techniques, and we are concerned that our current implementation is not sufficient to support a fair comparison.
>
> Q2. *Notation consistency/ambiguous*
> A2. Thank you for the suggestion! We are now aware that some notational inconsistencies and ambiguities are indeed present, and we will do a careful revision in the next version of the paper to ensure clarity.

---

> > ### Comment · Reviewer_qe9Q · 2025-08-01
> >
> > Thank you for your response. I have no further questions.

---

### Note · Authors · 2025-08-13

We sincerely thank the AC and all reviewers for their valuable feedback, constructive suggestions, and supportive comments that have helped us further improve the quality and clarity of our work. We greatly appreciate that the reviewers recognized the key strengths of our paper, including:

**Significance** – Distilling a large pre-trained flow-matching diffusion model into a few-step generator is an important and impactful research direction.

**Novelty** – Incorporating self-consistency loss inspired by shortcutting flow matching into the distillation process is a simple yet effective idea that aligns well with the underlying dynamics.

**Practicality** – Our method is evaluated on state-of-the-art text-to-image models, including SD3.5L and Flux1 Dev., and supports efficient distillation with LoRA, achieving few-step generation under only 24 A100 GPU hours.

**Strength of results** – The approach delivers strong performance, requires no architectural changes to the original model, and maintains low training cost.


We have carefully addressed all concerns raised during the review process. The reviewers’ supportive feedback has directly guided us in refining the manuscript, clarifying the motivation, and further highlighting the practical advantages and applicability of our method.

Finally, we are committed to releasing the full implementation and training code for our method upon acceptance, to facilitate transparency and reproducibility, and to benefit the research community.

We are grateful for the community’s encouragement and look forward to sharing this work with a broader audience.

---

### Decision · Program_Chairs · 2025-09-17

**Decision:**

Accept (poster)

**Comment:**

Tl;dr: Based on reviews, rebuttal and ensuing discussion I recommend accept.

### Paper summary

This paper introduces an efficient distillation method, termed ShortCut distillation for Flow Matching (SCFM) based pre-trained models. The core claim is obviating the need for explicit step-size conditioning. A new objective function based on the self-consistency principle is proposed. Reasonable empirical validation is provided, demonstrating that the method can distill very alrge models.

### Strengths and weaknesses

Strengths: 1) Efficiency: Both distillation time and data efficiency. 2) Results: Experimental results are appealing, 3) Simplicity: No model changes are required, results shown on two large models.

Weaknesses: 1) Theory: It is optimistically assumed that the velocity field will get straightened rather than explicitly being targeted or guaranteed. 2) As Reviewer VfPo notes, the existence of a single velocity field V(x_t, t) for all step sizes is not obvious. Author’s clarify this as an empirical observation. 3) Ablation: missing comparison with original shortcut model (R: SAAF) would make paper stronger.

### Decision justification

Main reasons are novelty, promising empirical results and practical significance. Important problem of slow sampling speed of diffusion models is tackled with a method that has low computational overhead. All three reviewers were satisfied with author responses and provided positive ratings.